# Clinical risk factors and blood protein biomarkers of 10-year pneumonia risk

Ming-Ming Lee[1], Yi Zuo[2], Katrina Steiling[3,4], Joseph P. Mizgerd[3], Bindu Kalesan[5], Allan J. Walkey[6]*

1 Pulmonary and Critical Care Medicine, Norwalk Hospital, Nuvance Health, Norwalk, CT, United States of America, 2 Department of Biostatistics, Vanderbilt University, Nashville, TN, United States of America, 3 The Pulmonary Center, Department of Medicine, Boston University School of Medicine, Boston, MA, United States of America, 4 Section of Computational Biomedicine, Boston University School of Medicine, Boston, MA, United States of America, 5 Boston University School of Medicine, Boston, MA, United States of America, 6 Division of Health Systems Science, Department of Medicine, UMass Chan Medical School, Worcester, MA, United States of America

* Allan.Walkey1@umassmed.edu

## Abstract

### Background

Chronic inflammation may increase susceptibility to pneumonia.

### Research question

To explore associations between clinical comorbidities, serum protein immunoassays, and long-term pneumonia risk.

### Methods

Framingham Heart Study Offspring Cohort participants ≥65 years were linked to their Centers for Medicare Services claims data. Clinical data and 88 serum protein immunoassays were evaluated for associations with 10-year incident pneumonia risk using Fine-Gray models for competing risks of death and least absolute shrinkage and selection operators for covariate selection.

### Results

We identified 1,370 participants with immunoassays and linkage to Medicare data. During 10 years of follow up, 428 (31%) participants had a pneumonia diagnosis. Chronic pulmonary disease [subdistribution hazard ratio (SHR) 1.87; 95% confidence interval (CI), 1.33–2.61], current smoking (SHR 1.79, CI 1.31–2.45), heart failure (SHR 1.74, CI 1.10–2.74), atrial fibrillation/flutter (SHR 1.43, CI 1.06–1.93), diabetes (SHR 1.36, CI 1.05–1.75), hospitalization within one year (SHR 1.34, CI 1.09–1.65), and age (SHR 1.06 per year, CI 1.04–1.08) were associated with pneumonia. Three baseline serum protein measurements were associated with pneumonia risk independent of measured clinical factors: growth differentiation factor 15 (SHR 1.32; CI 1.02–1.69), C-reactive protein (SHR 1.16, CI 1.06–1.27) and matrix metallopeptidase 8 (SHR 1.14, CI 1.01–1.30). Addition of C-reactive protein to the

**Data Availability Statement:** Our data comes from a merging of two data sources – the Framingham Heart Study Cohort and the United States Centers for Medicare and Medicaid Serves data. The authorship team do not have authorization to

distribute either data source separately or as a linked resource. We invite investigators interested in pursuing a dataset to contact the Framingham Heart Study for further information on procuring access to deidentified data from our study fhs@bu.edu.

**Funding:** The author(s) received no specific funding for this work.

**Competing interests:** The authors have declared that no competing interests exist.

clinical model improved prediction (Akaike information criterion 4950 from 4960; C-statistic of 0.64 from 0.62).

## Conclusions

Clinical comorbidities and serum immunoassays were predictive of pneumonia risk. C-reactive protein, a routinely-available measure of inflammation, modestly improved pneumonia risk prediction over clinical factors. Our findings support the hypothesis that prior inflammation may increase the risk of pneumonia.

## Introduction

Pneumonia is the leading infectious cause of death globally [1] and is a substantial public health issue. In the U.S., pneumonia accounts for an estimated 1.5 million adult hospitalizations annually with 100,000 in-hospital deaths [2]. Furthermore, pneumonia survivors experience numerous deleterious consequences, such as reduced pulmonary function, new cardiovascular complications, decline in cognitive function, depression, decreased functional status and quality of life, and shortened lifespans [3].

Prior studies have investigated clinical risk factors for pneumonia and found associations with age (both the youngest and oldest), chronic conditions, socioeconomic status and lifestyle factors such as smoking and alcohol ingestion [4]. Given associations between pneumonia, pre-existing comorbid conditions, and future health decline, some have hypothesized that pneumonia represents an acute manifestation of a chronic susceptibility state [5]. Although the mechanisms of increased pneumonia susceptibility are multi-factorial and incompletely understood, prior studies have posited links between chronic inflammation, immunosenescence and pneumonia risk [6–8]. However, few studies have evaluated associations between comorbidities, inflammation, and pneumonia in humans.

We explored the associations between clinical comorbidities, serum protein markers of organ function and inflammation, and long-term pneumonia risk in a well-characterized, community-dwelling, prospective cohort. Furthermore, we evaluated whether routinely-measured serum markers of inflammation such as C-reactive protein may be associated with future risk of pneumonia. Thus, we hypothesize that clinical risk factors and serum markers of inflammation would be associated with long-term risk for the development of pneumonia.

## Methods

### Data source and cohort

We conducted a retrospective cohort study including participants in the Framingham Heart Study (FHS) Offspring Cohort with FHS data linked to Centers for Medicare and Medicaid Services (CMS) claims data. The design of the FHS Offspring Cohort has been previously described [9], as well as linkage of FHS to CMS data [10]. In brief, FHS Offspring participants underwent standardized examinations and questionnaires with a study physician every 4 to 8 years. We used data from examinations 5 (1991–1995), 6 (1995–1998) and 7 (1998–2001) to characterize patient characteristics potentially associated with pneumonia based upon prior literature reviews [4]. Data missing from more recent examinations were carried forward if present at prior examinations. Data were missing if a participant did not answer a particular questionnaire item or the question was not asked at an examination. In addition to clinical

characteristics collected during examinations, FHS data contained measurements of 88 serum protein immunoassays drawn at examination 7 as part of the Systems Approach to Biomarker Research in Cardiovascular Disease Initiative, a sub-study of the FHS (see **S1 Table** in the online data supplement S1 File) [11]. Samples were obtained between 7 to 9 a.m. after an overnight fast and stored at -80˚C after centrifugation and until assay. Luminex™ xMAP multiplex immunoassay technology was used to measure the biomarkers [12]. Fourteen proteins were analyzed as binary variables defined as above or below a detection limit due to a high rate of missing data above the detection limit [11]. CMS inpatient, outpatient, skilled nursing facility, home health agency, hospice, carrier and durable medical equipment files were used to identify pneumonia diagnoses. Participants were included if they were 65 years old or older, had FHS data linked to CMS data, had no history of pneumonia prior to exam 7, and had a protein immunoassay panel measurement at examination 7 (**Fig 1**). Subjects were followed for up to 10 years from the time of examination 7 until diagnosis of pneumonia or death. Data for this study was de-identified, with a waiver of informed consent and protocol approval by the Boston University Medical Center Institutional Review Board (H-35870).

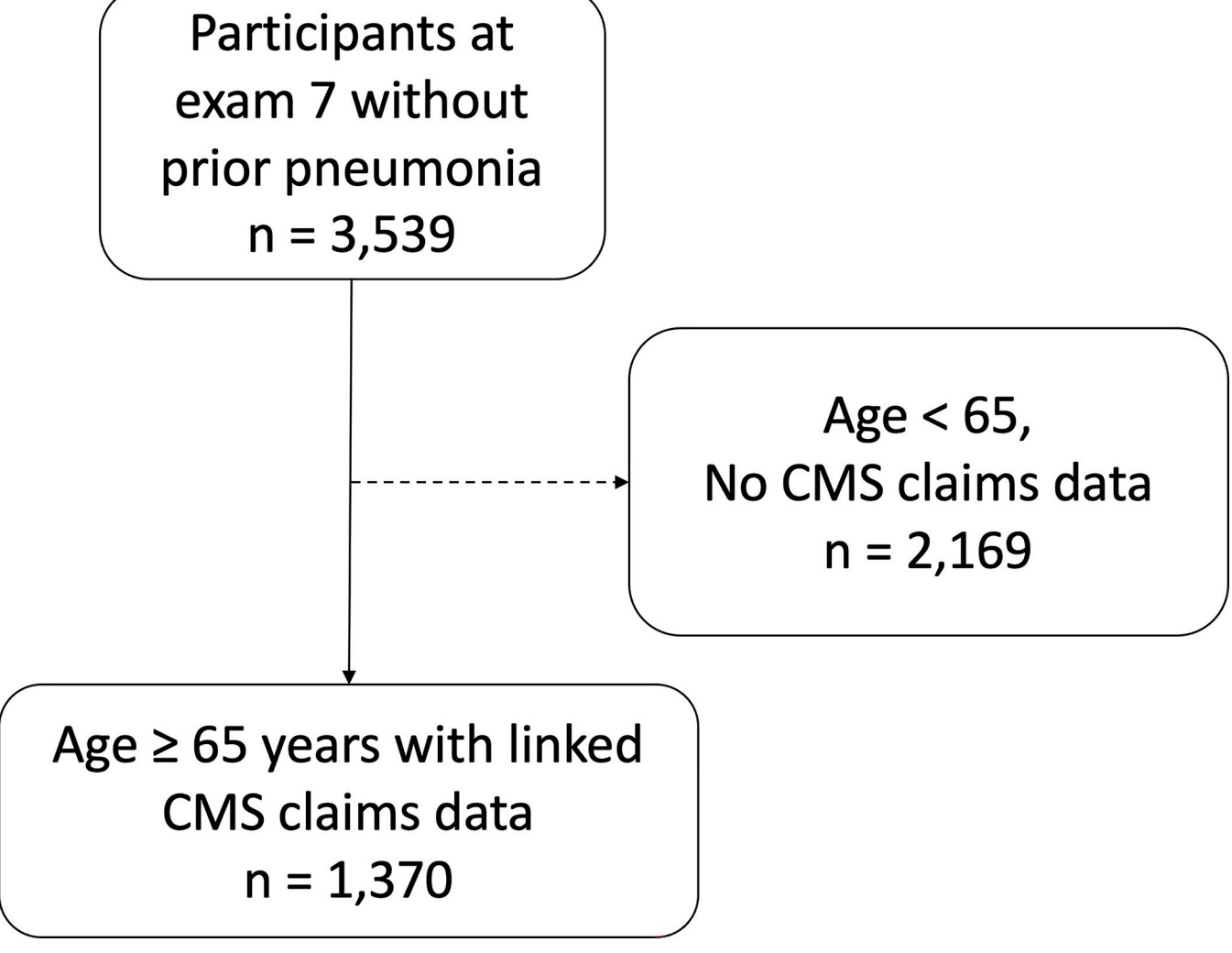

**Fig 1. Study enrollment.** Inclusion required participation in the Framingham Heart Study Exam 7. An age of less than 65 years at that visit or an absence of Centers for Medicare and Medicaid Services (CMS) data after that visit were criteria for exclusion.

## Outcomes

The primary outcome was pneumonia defined by International Classification of Diseases, Ninth Revision, Clinical Modification (ICD-9-CM) codes identified from CMS claims within 10 years of follow up. Based upon prior studies [13], we explored several ICD-9-CM coding algorithms for identifying acute pneumonia episodes from claims data. **S2 Table** in S1 File shows that clinical variables associated with pneumonia were similar regardless of the pneumonia definition based on various ICD-9CM codes for pneumonia and in various clinical settings, thus we chose to use the most sensitive definition capturing any pneumonia diagnosis in any clinical setting for all analyses. Death was defined from Medicare denominator files and loss to follow-up from Medicare files was likely rare.

## Covariates

We abstracted baseline demographics, clinical comorbidities, vital signs, anthropometric measurements (i.e., body mass index [BMI]), smoking and alcohol use and measures of functional and cognitive status from FHS clinical exam files. Clinical comorbidities included chronic pulmonary disease, hypertension, atrial fibrillation/flutter, valvular disease, myocardial infarction, heart failure, stroke, diabetes, chronic kidney disease, peripheral vascular disease, pulmonary embolus/deep vein thrombosis, cancer, hip fracture and depression as defined by FHS clinician assessments. Measures of functional and cognitive status included the Kannel physical activity index [14], mini-mental state examination [15], Katz index of independence in activities of daily living [16], physical disability [17] and overnight hospitalization in the year prior to the FHS examination. Detailed definitions of the covariates validated by the FHS protocols are provided in **S3 Table** in S1 File.

## Statistical methods

Cohort characteristics were compared as independent factors using Fisher's exact tests (for proportions) and unpaired t-tests (for means). We used a least absolute shrinkage and selection operator (LASSO) machine approach to select model covariates [18]. LASSO imposes a constraint to the model that minimizes prediction error and causes some regression coefficients to shrink toward 0, excluding those variables, and thus selects variables most strongly associated with the outcome of interest. Because of the presence of missing data deemed missing at random, we used a multiple imputation approach to missing data. We imputed missing data across 5 datasets and used LASSO methods to select covariates averaged across the pooled imputed data sets, where the tuning parameter lamba was selected using 5-fold cross-validation. Because of the potential for competing risk of death for pneumonia outcomes, we used Fine-Gray competing risk models to estimate the subdistribution hazard ratio of each selected predictor of pneumonia [19]. In order to explore contributions from comorbidities and serum protein markers to pneumonia risk, we estimated 4 risk models: 1) a model including only clinical variables from the FHS clinical exam data; 2) a model including only serum protein immunoassay measurements; 3) a model including both clinical variables and serum protein immunoassay measurements and 4) an exploratory model including clinical variables and routinely available protein biomarker C-reactive protein (CRP) (to assess how CRP may add to risk prediction above comorbidities). Models were assessed for fit by measured change in C-statistic (increase) and Akaike Information Criterion (AIC; decrease). Model calibration was assessed by plotting predicted risk of pneumonia against observed frequencies of pneumonia. Because optimal choice of model-building strategy is an area of debate [20] and strongly determines covariates included in the model, we performed sensitivity analyses building clinical models using multiple different strategies (e.g., p<0.2 based selection, use of Cox proportional

hazards models that did not account for competing risk of death) to assess the robustness of the multiple imputation LASSO selection approach (**S4 and S5 Tables** in S1 File). We performed additional sensitivity analyses using the four aforementioned risk models restricted to only pneumonia in the inpatient setting to assess for changes in model performance among patients with more severe pneumonia. All statistical analyses were performed with R software version 3.4.3.

## Results

### Cohort characteristics

We identified 1,370 FHS participants aged 65 years or older with linked CMS claims data and protein immunoassay panel measurements from examination 7. Subject characteristics are summarized in **Table 1**. During the 10 years of follow up, 428 (31%) subjects had a diagnosis of pneumonia.

Characteristics of subjects with and without a diagnosis of pneumonia are shown in **Table 2**. Patients with pneumonia more frequently were older, had higher rates of comorbid conditions (including atrial fibrillation, chronic kidney disease, chronic pulmonary disease, and heart failure), physical disability, smoking, and hospitalization within the prior year.

### Missing data

Of the 1,370 participants, the following data regarding baseline characteristics were missing: diabetes (n = 120 [9%]), peripheral vascular disease (n = 134 [10%]), Kannel physical activity index (n = 260 [19%]), valve disease (n = 134 [10%]), and BMI (n = 138 [10%]). WNK lysine deficient protein kinase 1 was excluded from analyses due to substantial missing data (n = 1,254 [92%]). Of the remaining 87 biomarkers, an average of 156 (11%) of samples were missing; fourteen biomarkers (**S1 Table** in S1 File), including interleukin-6 (IL-6), were analyzed as binary variables defined as above or below a detection limit due to missing data regarding levels above the detection limit.

### Factors associated with pneumonia

In the primary analysis using multiple imputation, Fine-Gray competing risk regression models, and LASSO variable selection, clinical variables associated with pneumonia were chronic pulmonary disease (subdistribution hazard ratio [SHR], 1.87; 95% CI, 1.33–2.61), current smoking (SHR, 1.79; 95% CI, 1.31–2.45), heart failure (SHR, 1.74; 95% CI, 1.10–2.74), atrial fibrillation/flutter (SHR, 1.43; 95% CI, 1.06–1.93), diabetes (SHR, 1.36; 95% CI, 1.05–1.75), hospitalization within one year (SHR, 1.34; 95% CI, 1.09–1.65), and age (SHR, 1.06; 95% CI, 1.04–1.08) (**Fig 2A**). In a model including only immunoassay measurements, five of 88 baseline serum protein measurements were predictive of pneumonia, including growth differentiation factor 15 (GDF-15; SHR, 1.65; 95% CI, 1.29–2.12), insulin-like growth factor binding protein 2 (IGFBP-2; SHR, 1.27; 95% CI, 1.00–1.60), matrix metallopeptidase 8 (MMP-8; SHR, 1.20; 95% CI, 1.06–1.37), C-reactive protein (CRP; SHR, 1.15; 95% CI, 1.04–1.27) and N-terminal prohormone of brain natriuretic peptide (NT-proBNP; SHR, 1.15; 95% CI, 1.04–1.28) (**Fig 2B**). In a model adjusted for clinical variables, baseline GDF-15 (SHR, 1.32; 95% CI, 1.02–1.69), MMP-8 (SHR, 1.14; 95% CI, 1.01–1.30) and CRP (SHR, 1.16; 95% CI, 1.06–1.27) remained significant predictors of pneumonia (**Fig 2C**). The addition of CRP (SHR, 1.17; 95% CI, 1.06–1.28) to the clinical model improved the clinical model fit (AIC value 4960 to 4950) and discrimination of pneumonia events (C-statistic 0.62 to 0.64) (**Fig 2D**). Calibration plots showed good calibration across the models, except at extreme levels of pneumonia risk, with

**Table 1. Baseline characteristics of participants (N = 1,370).**

| | |
|---|---|
| **Demographics** | |
| Age, years | 71.5 ± 4.8 |
| Male | 631 (46.1) |
| Married | 960 (70.1) |
| Residence place | |
| Private residence | 1,284 (93.7) |
| Nursing home | 18 (1.3) |
| Other institution | 55 (4.0) |
| **Substance use history** | |
| Alcohol use | 125 (9.1) |
| Current smoker | 115 (8.4) |
| **Clinical comorbidities** | |
| Hypertension | 868 (63.4) |
| Depression | 362 (26.4) |
| Chronic kidney disease | 282 (20.6) |
| Cancer | 253 (18.5) |
| Valve disease | 242 (17.7) |
| Diabetes | 222 (16.2) |
| Atrial fibrillation/flutter | 160 (11.7) |
| Myocardial infarction | 132 (9.6) |
| Chronic pulmonary disease | 86 (6.3) |
| Peripheral vascular disease | 67 (4.9) |
| Heart failure | 55 (4.0) |
| Hip fracture | 31 (2.3) |
| Deep vein thrombosis | 14 (1.0) |
| Stroke | 14 (1.0) |
| Pulmonary embolus | 5 (0.4) |
| **Cognitive and functional status** | |
| Mini-mental state exam | 27.7 ± 3.8 |
| Kannel physical activity index | 38.4 ± 6.3 |
| Activities of daily living score | |
| 0 | 21 (1.5) |
| 1 | 10 (0.7) |
| 2 | 5 (0.4) |
| 3 | 8 (0.6) |
| 4 | 20 (1.5) |
| 5 | 1,306 (95.3) |
| Physical disability | 90 (6.6) |
| Overnight hospitalization in the past year | |
| Yes, once | 290 (21.2) |
| Yes, more than once | 136 (9.9) |
| No | 943 (68.8) |
| **Other measures** | |
| Heart rate, beats per minute | 64.9 ± 11.5 |
| Mean arterial pressure, mmHg | 92.1 ± 11.2 |
| Body mass index category | |
| Underweight (<18.5) | 9 (0.7) |
| Normal (18.5–24.9) | 358 (26.1) |

*(Continued)*

**Table 1.** (Continued)

| | |
|---|---|
| Overweight (25–29.9) | 524 (38.3) |
| Obese (30–39.9) | 309 (22.6) |
| Morbid obesity (≥40) | 32 (2.3) |

*Definition of abbreviation*: SD = standard deviation.

Data presented as mean ± SD or n (%).

the exploratory clinical and CRP model showing optimal calibration (**S1 Fig** in S1 File). Sensitivity analyses of clinical variables in complete case analysis showed similar results to the multiple imputed LASSO approach (**S6 Table** in S1 File). Sensitivity analyses using the risk models restricted to pneumonia in the inpatient setting did not substantively change model performance (C-statistic of the model including only clinical variables was 0.67; C-statistic of the model including only serum protein immunoassays was 0.64; C-statistic of the model including clinical variables and serum protein immunoassays was 0.69; C-statistic of the exploratory model including clinical covariates and C-reactive protein was 0.68 (**S7 Table** in S1 File).

## Discussion

We explored clinical and immunoassay predictors of pneumonia risk in a cohort of community dwelling participants in the FHS. In addition to observing multiple clinical variables associated with pneumonia, we identified 3 baseline serum protein immunoassay measures (GDF-15, MMP-8 and CRP), after adjusting for clinical variables, that were associated with pneumonia events up to 10 years prior to the pneumonia diagnosis. Furthermore, the addition of routinely-available CRP to a model including clinical variables improved model performance for discrimination of pneumonia events. Models performed similarly when restricted to more severe pneumonia requiring hospitalization. Our findings suggest that long-term pneumonia risks are associated not only with previously described clinical characteristics, but also with serum markers of inflammation. Taken together, our findings provide preliminary support to the hypothesis that prior inflammation may be a marker of increased pneumonia risk and inform future efforts at pneumonia risk prediction.

We caution against interpreting the association between the clinical variables and protein immunoassays and future pneumonia as casual and acknowledge that these predictors may correlate with unmeasured causal variables. In addition, single measurements of protein immunoassays may not represent a chronic inflammatory state. However, our results expand upon prior studies of inflammation, clinical comorbidities, and pneumonia risk. Yende et al. [8] identified associations between increased baseline levels of inflammatory cytokines (IL-6 and tumor necrosis factor-alpha [TNF-α]) and future pneumonia risk in a cohort of well-functioning, community-dwelling elderly subjects. Unfortunately, a high rate of missing data in our study restricted analysis of IL-6, and tumor necrosis factor-alpha was not included in the immunoassay panel. However, we found other markers of inflammation independent of clinical comorbidities associated with long-term pneumonia risk, including routinely-available clinical measures such as CRP, an acute-phase protein synthesized by the liver. Although the utility of CRP in the diagnosis of acute pneumonia and as a predictor of more severe disease with poor outcomes has been reported [21], the association of baseline CRP with future pneumonia risk has not been described.

Inflammation may increase pneumonia susceptibility through various biological mechanisms. The highest incidence of pneumonia in adults is amongst the elderly, and aging is

**Table 2. Distribution of baseline characteristics by pneumonia outcome (N = 1,370).**

| | Any pneumonia | No pneumonia | P |
| --- | --- | --- | --- |
| | **(n = 428)** | **(n = 942)** | **(* if significant)** |
| **Demographics** | | | |
| Age, years | 72.8 ± 5.0 | 70.9 ± 4.6 | <0.0001 * |
| Male | 207 (48.4) | 424 (45.0) | 0.27 |
| Married | 301 (70.3) | 659 (70.0) | 0.90 |
| Residence place | | | |
| Private residence | 400 (94.8) | 884 (93.6) | 0.81 |
| Nursing home | 6 (1.4) | 12 (1.3) | 0.80 |
| Other institution | 16 (3.8) | 39 (4.2) | 0.77 |
| Current smoker | 52 (12.2) | 63 (6.7) | 0.0011 * |
| Alcohol use | 36 (8.4) | 89 (9.4) | 0.61 |
| **Clinical comorbidities** | | | |
| Atrial fibrillation/flutter | 77 (18.0) | 83 (8.8) | <0.0001 * |
| Cancer | 85 (19.9) | 168 (17.8) | 0.37 |
| Chronic kidney disease | 116 (27.1) | 166 (17.6) | <0.0001 * |
| Chronic pulmonary disease | 45 (10.5) | 41 (4.4) | <0.0001 * |
| Depression | 125 (29.2) | 237 (25.2) | 0.13 |
| Diabetes | 86 (20.1) | 136 (14.4) | 0.011 |
| Heart failure | 32 (7.5) | 23 (2.4) | <0.0001 * |
| Hip fracture | 15 (3.5) | 16 (1.7) | 0.049 |
| Hypertension | 292 (68.2) | 576 (61.2) | 0.013 |
| Peripheral vascular disease | 28 (6.5) | 39 (4.1) | 0.059 |
| Pulmonary embolus | 4 (0.9) | 1 (0.1) | 0.036 |
| Deep vein thrombosis | 6 (1.4) | 8 (0.9) | 0.39 |
| Myocardial infarction | 43 (10.1) | 89 (9.5) | 0.77 |
| Stroke | 6 (1.5) | 8 (0.8) | 0.39 |
| Valvular disease | 82 (19.2) | 160 (17.0) | 0.36 |
| **Cognitive and functional status** | | | |
| Mini-mental state exam | 27.5 ± 3.9 | 27.8 ± 3.8 | 0.18 |
| Kannel physical activity index | 37.9 ± 6.1 | 38.6 ± 6.4 | 0.057 |
| Activities of daily living score | | | |
| 0 | 7 (1.6) | 14 (1.5) | 0.82 |
| 1 | 1 (0.2) | 9 (1.0) | 0.19 |
| 2 | 2 (0.5) | 3 (0.3) | 0.65 |
| 3 | 2 (0.5) | 6 (0.6) | >0.99 |
| 4 | 10 (2.3) | 10 (1.1) | .087 |
| 5 | 406 (94.9) | 900 (95.5) | 0.58 |
| Physical disability | 44 (10.3) | 46 (4.9) | 0.0004 * |
| Overnight hospitalization in the past year | | | |
| Yes, once | 113 (26.5) | 177 (18.8) | 0.0017 |
| Yes, more than once | 52 (12.2) | 84 (8.9) | 0.079 |
| No | 262 (61.4) | 681 (72.3) | <0.0001 * |
| **Other measures** | | | |
| Heart rate, beats per minute | 65.2 ± 11.8 | 64.8 ±11.3 | 0.55 |
| Mean arterial pressure, mmHg | 91.4 ± 11.4 | 92.5 ± 11.1 | 0.092 |
| Body mass index category | | | |
| < 18.5 underweight | 5 (1.4) | 4 (0.5) | 0.15 |

*(Continued)*

**Table 2.** (Continued)

|  | **Any pneumonia** | **No pneumonia** | *P* |
|---|---|---|---|
|  | **(n = 428)** | **(n = 942)** | **(* if significant)** |
| 18.5–24.9 normal | 115 (31.0) | 243 (28.2) | 0.69 |
| 25–29.9 overweight | 137 (36.9) | 387 (45.0) | 0.0015 |
| 30–39.9 obese | 101 (27.2) | 208 (24.2) | 0.53 |
| > = 40 morbid obesity | 13 (3.5) | 19 (2.2) | 0.25 |

*Definition of abbreviation*: SD = standard deviation.

Data presented as mean ± SD or n (%). *P* values report unadjusted probabilities from group comparisons using unpaired t-tests (for values reported as mean with SD) or Fisher's exact tests (for values reported as n with %), with asterisks (*) indicating statistical significance for a false discovery rate <0.05 using the Bonferroni correction for multiple comparisons.

associated with inflammation and immunosenescence [22, 23]. Among older adults, those with frailty are more likely to experience pneumonia [24, 25]; frailty also associates with systemic inflammation [26, 27] including serum CRP in the cohort here studied [28]. Aged lungs experience greater pathogen adhesion and host cell invasion, a delayed and diminished immune response during early infection, attenuated phagocytic capabilities of macrophages and neutrophils via Toll-like receptor dysfunction, and a prolonged immune response to infection [6, 7, 29, 30]. These pre-clinical findings suggest age-related chronic inflammation results in immune dysregulation that may contribute to increased pneumonia susceptibility. But, the exact causal pathways between the clinical factors and biomarkers and future pneumonia risk remain unclear.

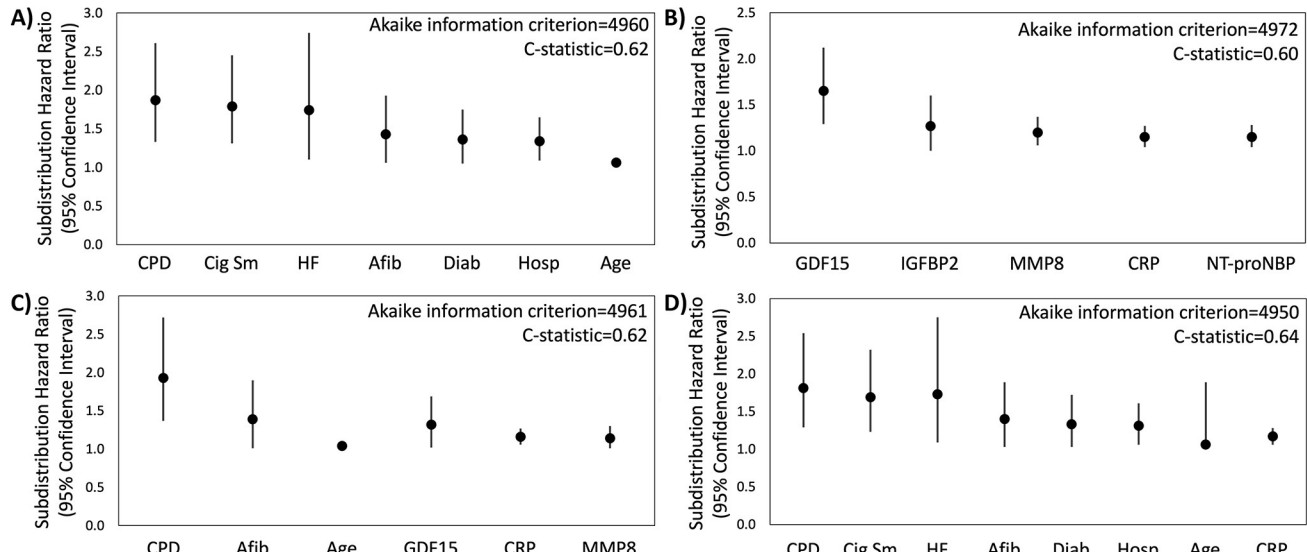

**Fig 2. Clinical variables and immunoassay measurements as predictors of subsequent pneumonia.** Subdistribution Hazard Ratios are shown for factors reaching statistical significance, as well as model fit estimates using the Akaike information criterion and C-statistic. Immunoassay values were log-transformed prior to analyses. A) In a model including clinical variables alone, chronic pulmonary disease (CPD), current smoking (Cig Sm), heart failure (HF), atrial fibrillation/flutter (Afib), diabetes (Diab), hospitalization within one year (Hosp), and age were predictive of pneumonia. B) In a model including immunoassay measurements alone, growth differentiation factor 15 (GDF15), insulin-like growth factor binding protein 2 (IGFBP2), matrix metallopeptidase 8 (MMP8), C-reactive protein (CRP), and N-terminal prohormone of brain natriuretic peptide (NT-proBNP) were predictive of pneumonia. C) After adjusting for clinical variables, GDF-15, MMP-8, and CRP remained significant. D) Adding CRP to the model based on clinical variables alone (depicted in panel A) reduced the AIC value (showing improved clinical model fit) and increased the C-statistic (showing better discrimination of pneumonia events).

In addition to CRP, baseline GDF-15 and MMP-8 were also associated with future pneumonia risk independent of clinical comorbidities. The association of these inflammatory markers with future pneumonia risk has not been previously reported. Although our methodological approach was structured to maximize pneumonia risk prediction, and not establish causal pathways, we review below previously described biological roles of the biomarkers associated with pneumonia risk to guide potential future research. GDF-15 is a transforming growth factor-ß superfamily cytokine that is expressed in various cell types, and its expression increases in response to tissue injury and inflammation [31]. In subjects with chronic obstructive pulmonary disease (COPD) and in response to cigarette smoke exposure, GDF-15 is upregulated in airway epithelial cells and activates pathways that promote mucin production and cellular senescence, altering mucosal immunity and leading to chronic airway inflammation, respectively. Furthermore, GDF-15 overexpression has been shown to increase susceptibility to and severity of respiratory viral infections via increased viral replication and exaggerated virus-induced inflammation [32]. Baseline serum GDF-15 levels have been found to be an independent predictor of adverse outcomes in COPD patients, including accelerated pulmonary function decline, higher yearly exacerbation rates and increased mortality [33]. Finally, increased GDF-15 expression in the muscle of COPD subjects suggests it contributes to muscle wasting in COPD [34]. Although these studies report the association of GDF-15 with increased susceptibility to viral infection, chronic inflammation and senescence in the context of COPD and cigarette smoke exposure, they introduce possible mechanisms by which baseline GDF-15 increases future pneumonia risk. MMP-8 is a member of the matrix metalloproteinase family of proteases and is involved in the breakdown of extracellular matrix and cleavage of chemokines [35]. MMPs contribute to the pathogenesis of multiple inflammatory pulmonary conditions such as emphysema, COPD, bronchiectasis and acute respiratory distress syndrome through primarily increased neutrophilic activity [36–39]. Hartog et al. [40] found elevated levels of MMP-8 in the bronchoalveolar lavage fluid of patients with hospital-acquired pneumonia, with levels correlated to clinical severity. Our findings add to Hartog et al.'s by demonstrating that baseline MMP-8 levels may predict future pneumonia. Potential mechanisms linking baseline levels of CRP, GDF-15 and MMP-8 to pneumonia risk require further study.

Our study also examined the association of clinical comorbidities and lifestyle factors with pneumonia susceptibility, and our finding that age, atrial fibrillation, chronic pulmonary disease, diabetes, heart failure, prior overnight hospitalization and smoking were associated with pneumonia risk is similar to that of previous studies [4, 23, 41] providing face validity to our results. In contrast to prior studies [4, 42], we did not observe chronic renal disease, cancer, BMI, alcohol use and nursing home residence to be independently predictive of future pneumonia risk. The adjustment for additional clinical covariates and potential confounders in our study as compared with prior studies, which may be partially collinear with some comorbidities, as well as the use of competing risk statistical models, may explain some of the differences between our results and prior studies of pneumonia risk factors. Despite studies showing an association between cognitive impairment and pneumonia [4, 43], we interestingly did not find mini-mental status scores to be predictive of future pneumonia. Regardless, clinical comorbidities increase pneumonia susceptibility through a variety of mechanisms, in addition to chronic inflammation, that increase pathogen burden and impair immune responses. A more detailed description of links between clinical comorbidities and pneumonia risk can be found in Quinton et al. [3].

Our study has several strengths and limitations that bear consideration. Our current prediction model is not intended to be used for clinical decision-making, however it highlights a novel association between inflammatory markers and future pneumonia. We chose to study FHS participants because they are a cohort of well characterized, community-dwelling

residents with available serum protein immunoassay measurements and linked CMS data; the panel of 88 immunoassays performed on over a thousand subjects in the community provides unusually deep insights into serum inflammatory markers. However, we acknowledge the lack of generalizability of our findings to other dissimilar populations. Thus, our findings of associations between individual immunoassays and pneumonia risk are hypothesis-generating and should be replicated in additional datasets prior to further exploration of potential mechanisms between these biomarkers and pneumonia susceptibility. Associations with subsequent pneumonias were assessed using data from blood collections during a single study visit. Because this provides a snapshot of biological and disease processes that are dynamic, the degree to which these protein profiles and clinical information were stable representations of these individuals is uncertain. Thus, risk factors for pneumonia may be refined with future studies that include serial assessments of blood proteins and co-morbidities. The decade of follow-up available for this cohort proved helpful for capturing future pneumonia cases amongst subjects who are 65 years or older, achieving a prevalence of 31% across that time-frame (428 of the 1,370 subjects). Furthermore, because we used the most sensitive definition of pneumonia for analysis, we found biomarkers that predicted future pneumonia of various types (i.e., community-acquired, hospital-acquired, bacterial, viral, etc.). However, the CMS data were not sufficient to stratify those pneumonia types, and we lacked data on the pneumonia-causing pathogens. Thus, future studies examining more specific definitions of pneumonia and causative pathogens might improve our understanding of the pathogenesis of different types of pneumonia. We also did not include the effect of vaccination, such as vaccination against pneumococcus or influenza, immunosuppressive medications, other infections or time-varying factors on pneumonia risk. In addition, studies have shown that the microbiome plays an important role in host immunity and that dysbiosis leads to impaired immune defenses [44] and potentially chronic inflammation. Future studies should explore potential relationships between dysbiosis, inflammation and pneumonia risk. Missing data on specific biomarkers (e.g., IL-6), use of multiple imputation and multiple hypothesis tests may increase risk of both systematic and random bias. However, we used methods designed to minimize bias such as multiple imputation, and LASSO covariate selection, and multiple sensitivity analyses demonstrated robustness of our models. Finally, misclassification of pneumonia diagnoses by ICD-9-CM codes may increase risk of bias.

In conclusion, we found clinical and serum protein predictors of pneumonia risk. Our findings showing associations between biomarkers of inflammation and pneumonia support the hypothesis that pneumonia is, in part, a disease of persistent or increased susceptibility. The exact mechanisms by which aging, inflammation and clinical comorbidities increase pneumonia susceptibility remain fertile areas for further investigation and potential targets for novel interventions to reduce pneumonia risk.

## Supporting information

**S1 File.**
(DOCX)

## Acknowledgments

M.L., A.J.W., Y.Z. and B.K. are responsible for the content of the manuscript, including the integrity of the data and the accuracy of analysis.

None of the authors received direct funding for the study. The study data resources were supported by the National Heart, Lung, and Blood Institute (NHLBI) in collaboration with

Boston University (Contract No. N01-HC-25195 and HHSN268201500001I). Additional funding for SABRe immunoplex panel was provided by Division of Intramural Research, NHLBI, and Center for Population Studies, NHLBI to the Framingham Heart Study (FHS) investigators, who are not authors of the current manuscript. Funding support for the Framingham Multiplexed Immunoassay Panel 1–17 (SABRe project) datasets was provided by FHS contract supplement, NHLBI Intramural Funding. This manuscript was not prepared in collaboration with investigators of the FHS and does not necessarily reflect the opinions or views of the FHS, Boston University, or NHLBI.

## Author Contributions

**Conceptualization:** Joseph P. Mizgerd, Bindu Kalesan, Allan J. Walkey.

**Data curation:** Yi Zuo, Bindu Kalesan, Allan J. Walkey.

**Formal analysis:** Yi Zuo, Bindu Kalesan.

**Investigation:** Allan J. Walkey.

**Methodology:** Ming-Ming Lee, Katrina Steiling, Bindu Kalesan, Allan J. Walkey.

**Project administration:** Allan J. Walkey.

**Supervision:** Allan J. Walkey.

**Writing – original draft:** Ming-Ming Lee.

**Writing – review & editing:** Yi Zuo, Katrina Steiling, Joseph P. Mizgerd, Bindu Kalesan, Allan J. Walkey.

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
