## [Decision Letter · Decision Letter 0]

3 Apr 2024

PONE-D-23-40628Clinical risk factors and blood protein biomarkers of 10-year pneumonia riskPLOS ONE

Dear Dr. Walkey,

Thank you for submitting your manuscript to PLOS ONE. After careful consideration, we feel that it has merit but does not fully meet PLOS ONE’s publication criteria as it currently stands. Therefore, we invite you to submit a revised version of the manuscript that addresses the points raised during the review process.

We look forward to receiving your revised manuscript.

Kind regards,

Gaetano Santulli

Academic Editor

PLOS ONE

Reviewers' comments:

Reviewer's Responses to Questions

**Comments to the Author**

1. Is the manuscript technically sound, and do the data support the conclusions?

Reviewer #1: Partly

Reviewer #2: Partly

2. Has the statistical analysis been performed appropriately and rigorously? 

Reviewer #1: Yes

Reviewer #2: Yes

3. Have the authors made all data underlying the findings in their manuscript fully available?

Reviewer #1: Yes

Reviewer #2: Yes

4. Is the manuscript presented in an intelligible fashion and written in standard English?

Reviewer #1: Yes

Reviewer #2: Yes

5. Review Comments to the Author

Reviewer #1: This study integrated the serum immunoassays and clinical comorbidities from Framingham Heart Study Offspring Cohort participants to predict their 10-year incident pneumonia risk. This is a well-written article, while I have some concerns of this study.

1. Though the participants in this study are most from community, pneumonia is a sporadic infectious disease; this study can’t provide the pathogens to know the potential transmission mode. Additionally, the immune biomarkers are dynamic depending on individual’s healthy status, thus the association between biomarkers prior 10 years and incidental event is controversial.

2. The immunity profiles are associated with participants’ underlying diseases. The authors should provide p-value to demonstrate the difference between 2 groups.

According to the Table 2 “ Distribution of baseline characteristics by pneumonia outcome” the pneumonia group seems to have high ratio of cancer. Chronic kidney disease, chronic pulmonary diseases, heart failure, frequency of overnight hospitalization in the past year…etc, which indicates the pneumonia group may have under more inflammation status than nonpneumonia group and may encounter more illness events in the future.

3. This study used LASSO machine approach to select model covarients with 5-fold cross validation and established 4 models to show the performance of pneumonia prediction. I suggest that the authors may randomly divided these 1370 participants into training cohort for model construction and use others for independent validation to show the performance of best predictive model.

Reviewer #2: The manuscript described the clinical risk factors and blood protein biomarkers of 10-year pneumonia risk. I found the manuscript of some interest with some points to improve. First of all, age is a key factor in your population, please discuss the role of frailty and aging in your results:

doi: 10.1371/journal.pone.0300898.

doi: 10.3389/fragi.2023.1345486.

doi: 10.1016/j.ejim.2023.05.027.

doi: 10.1111/jch.14439.

doi: 10.1371/journal.pone.0296014.

Figures must be increases in font size. Also, increase the resolution and re-submit in TIFF.

Still, Figure captions should be provided.

The limitations and the strengths should be better detailed.

6. PLOS authors have the option to publish the peer review history of their article (what does this mean?). If published, this will include your full peer review and any attached files.

Reviewer #1: No

Reviewer #2: No

---

## [Author Response · Author response to Decision Letter 0]

25 Apr 2024

Dear Editor,

Thank you for soliciting the constructive reviews of our manuscript. We offer a point-by-point response below. We believe that the revised manuscript is improved.

Regards, 

Allan J. Walkey, MD for the authorship team

Editorial comment 1. When submitting your revision, we need you to address these additional requirements.

Response to Editors1. We revised extensively to meet PLOS ONE style requirements.

E2. We note that you have indicated that there are restrictions to data sharing for this study. PLOS only allows data to be available upon request if there are legal or ethical restrictions on sharing data publicly. For more information on unacceptable data access restrictions, please see http://journals.plos.org/plosone/s/data-availability#loc-unacceptable-data-access-restrictions. 

RE2. Our data comes from a merging of two data sources – the Framingham Heart Study Cohort and the United States Centers for Medicare and Medicaid Serves data. The authorship team do not have authorization to distribute either data source separately or as a linked resource. We invite investigators interested in pursuing a dataset to contact the Framingham Heart Study for further information on procuring access to deidentified data from our study fhs@bu.edu. 

E3. Please include captions for your Supporting Information files at the end of your manuscript, and update any in-text citations to match accordingly. Please see our Supporting Information guidelines for more information: http://journals.plos.org/plosone/s/supporting-information. 

RE3. Captions were added at the end for the Supporting Information files, and in-text citations were matched accordingly.

Reviewer comments

Reviewer #1: This study integrated the serum immunoassays and clinical comorbidities from Framingham Heart Study Offspring Cohort participants to predict their 10-year incident pneumonia risk. This is a well-written article, while I have some concerns of this study.

Query1. Though the participants in this study are most from community, pneumonia is a sporadic infectious disease; this study can’t provide the pathogens to know the potential transmission mode. Additionally, the immune biomarkers are dynamic depending on individual’s healthy status, thus the association between biomarkers prior 10 years and incidental event is controversial.

Response 1. The reviewer states the thrust of the current study and highlights some limitations. Our goal with this study was to compare and contrast individuals who later develop pneumonia after a study visit to those who do not, in order to identify elements (including but not limited to biomarkers) that may be predictive indicators of pneumonia susceptibility. 

Pneumonia is a complex syndrome which includes many different types of pathogens, hosts, and host-pathogen interactions, yielding a wide variety of divergent pathophysiologies (e.g., see DOI: 10.1152/physrev.00032.2017). Microbial etiology is usually unknown for pneumonia cases, even when intensively investigated in hospitalized patients using aggressive patient sampling and microbial analyses (e.g., see DOI: 10.1056/NEJMoa1500245). The subjects in this study were not enrolled in clinical studies, were not all hospitalized, and were not aggressively sampled or analyzed during pneumonia. Only routine clinical information is available from these subjects, and microbial etiology cannot be inferred. We communicate this limitation in the second-to-last paragraph of the Discussion: “…we found biomarkers that predicted future pneumonia of various types (i.e., community-acquired, hospital-acquired, bacterial, viral, etc.). However, the CMS data were not sufficient to stratify those pneumonia types, and we lacked data on the pneumonia-causing pathogens. Thus, future studies examining more specific definitions of pneumonia and causative pathogens might improve our understanding of the pathogenesis of different types of pneumonia.” 

The reviewer correctly notes that our study relied on a single visit to collect blood samples and assess co-morbidities. Although we had acknowledged the concern (“single measurements of protein immunoassays may not represent a chronic inflammatory state” in paragraph 2 of the Discussion), we had not sufficiently emphasized this limitation in our original submission. We corrected this by revising the Discussion section to add the following sentences in the second-to-last paragraph: 

Manuscript Change, Discussion:

“Associations with subsequent pneumonias were assessed using data from blood collections and clinical assessments during a single study visit. Because this provides a snapshot of biological and disease processes that are dynamic, the degrees to which these protein profiles and clinical information were stable representations for these individuals is uncertain. Thus, risk factors for pneumonia may be refined with future studies that include serial assessments of blood proteins and co-morbidities.” 

Q2. The immunity profiles are associated with participants’ underlying diseases. The authors should provide p-value to demonstrate the difference between 2 groups.

According to the Table 2 “ Distribution of baseline characteristics by pneumonia outcome” the pneumonia group seems to have high ratio of cancer. Chronic kidney disease, chronic pulmonary diseases, heart failure, frequency of overnight hospitalization in the past year…etc, which indicates the pneumonia group may have under more inflammation status than nonpneumonia group and may encounter more illness events in the future.

R2. As suggested by the reviewer, we collected P values for each cohort characteristic in Table 2. The statistically significant differences (after correcting for multiple comparisons) between the groups that did vs. did not later develop pneumonia were age, smoking status, atrial fibrillation, chronic kidney disease, chronic pulmonary disease, heart failure, physical disability, and hospitalization within the prior year. We revised the Results section under “Cohort Characteristics” to highlight these findings. We also revised Table 2 to show all P values, as suggested.

Q3. This study used LASSO machine approach to select model covarients with 5-fold cross validation and established 4 models to show the performance of pneumonia prediction. I suggest that the authors may randomly divided these 1370 participants into training cohort for model construction and use others for independent validation to show the performance of best predictive model.

R3. The reviewer suggests a single derivation-validation cohort split be used rather than the 5-fold cross validation approach used in our original analysis. Note, in neither approach do we have a validation set from an independent data source. Thus, both approaches randomly sample data from among the 1370 participants, holding out some data from initial test model development for later model validation. The 5-fold cross validation uses 5 hold-out data groups to evaluate the performance of the model and pools the average performance across the 5 different validation models to determine model performance, whereas the single derivation-validation sets proposed by the reviewer only holds out one data group for validation, and is identical to a 2-fold cross validation. Because 5-fold cross validation results in more accurate results than the 2-fold cross validation (see for example, Zhang P. Model selection via multifold cross validation. The Annals of Statistics. 1993;21(1):299–313) suggested by the reviewer, we have maintained the 5-fold cross validation approach. 

Reviewer #2, 

Q4: The manuscript described the clinical risk factors and blood protein biomarkers of 10-year pneumonia risk. I found the manuscript of some interest with some points to improve. First of all, age is a key factor in your population, please discuss the role of frailty and aging in your results:

doi: 10.1371/journal.pone.0300898.

doi: 10.3389/fragi.2023.1345486.

doi: 10.1016/j.ejim.2023.05.027.

doi: 10.1111/jch.14439.

doi: 10.1371/journal.pone.0296014.

R4. We agree that frailty may be one of multiple factors that contribute to older individuals being more prone to pneumonia. The references about frailty and aging provided by the reviewer were helpful, but they did not connect directly to pneumonia or to the inflammation markers measured in our study. However, this suggestion from the reviewer led to manuscript revisions presenting known relationships of frailty to chronic inflammation (including CRP measurements) and pneumonia. Frailty increases the risk of pneumonia for older individuals. We revised the Discussion to communicate that frailty may contribute to the pneumonia risk in our older subjects, in the paragraph about aging as a risk factor for pneumonia in the Discussion, as suggested by the reviewer.

Manuscript change, Discussion.

“Among older adults, those with frailty are more likely to experience pneumonia [doi:10.1038/s41598-021-86854-3, doi:10.1186/s12877-023-03979-y]; frailty also associates with systemic inflammation [DOI: 10.1038/s41569-018-0064-2, doi:10.1016/j.exger.2023.112253] including serum CRP in the cohort here studied [DOI: 10.1007/s11357-015-9864-z].”

Q5. Figures must be increases in font size. Also, increase the resolution and re-submit in TIFF.

R5. Revised as suggested. Font size was increased in the figures. High resolution TIFF versions were generated for the revised submission.

Q6. Still, Figure captions should be provided.

R6. Revised as suggested. We added legends for both figures, and placed them where instructed by the PLOS ONE style guidelines.

Q7. The limitations and the strengths should be better detailed.

R7. Revised as suggested. The second-to-last paragraph was expanded to better detail strengths and limitations of the of the current study.

Manuscript Change, Discussion

“Our study has several strengths and limitations that bear consideration. Our current prediction model is not intended to be used for clinical decision-making, however it highlights a novel association between inflammatory markers and future pneumonia. We chose to study FHS participants because they are a cohort of well characterized, community-dwelling residents with available serum protein immunoassay measurements and linked CMS data; the panel of 88 immunoassays performed on over a thousand subjects in the community provides unusually deep insights into serum inflammatory markers. However, we acknowledge the lack of generalizability of our findings to other dissimilar populations. Thus, our findings of associations between individual immunoassays and pneumonia risk are hypothesis-generating and should be replicated in additional datasets prior to further exploration of potential mechanisms between these biomarkers and pneumonia susceptibility. Associations with subsequent pneumonias were assessed using data from blood collections during a single study visit. Because this provides a snapshot of biological and disease processes that are dynamic, the degree to which these protein profiles and clinical information were stable representations of these individuals is uncertain. Thus, risk factors for pneumonia may be refined with future studies that include serial assessments of blood proteins and co-morbidities. The decade of follow-up available for this cohort proved helpful for capturing future pneumonia cases amongst subjects who are 65 years or older, achieving a prevalence of 31% across that time-frame (428 of the 1,370 subjects). Furthermore, because we used the most sensitive definition of pneumonia for analysis, we found biomarkers that predicted future pneumonia of various types (i.e., community-acquired, hospital-acquired, bacterial, viral, etc.). However, the CMS data were not sufficient to stratify those pneumonia types, and we lacked data on the pneumonia-causing pathogens. Thus, future studies examining more specific definitions of pneumonia and causative pathogens might improve our understanding of the pathogenesis of different types of pneumonia. We also did not include the effect of vaccination, such as vaccination against pneumococcus or influenza, immunosuppressive medications, other infections or time-varying factors on pneumonia risk. In addition, studies have shown that the microbiome plays an important role in host immunity and that dysbiosis leads to impaired immune defenses [39] and potentially chronic inflammation. Future studies should explore potential relationships between dysbiosis, inflammation and pneumonia risk. Missing data on specific biomarkers (e.g., IL-6), use of multiple imputation and multiple hypothesis tests may increase risk of both systematic and random bias. However, we used methods designed to minimize bias such as multiple imputation, and LASSO covariate selection, and multiple sensitivity analyses demonstrated robustness of our models. Finally, misclassification of pneumonia diagnoses by ICD-9-CM codes may increase risk of bias. “

---

## [Editor Report · Decision Letter 1]

29 Apr 2024

Clinical risk factors and blood protein biomarkers of 10-year pneumonia risk

PONE-D-23-40628R1

Dear Dr. Walkey,

We’re pleased to inform you that your manuscript has been judged scientifically suitable for publication and will be formally accepted for publication once it meets all outstanding technical requirements.

Kind regards,

Gaetano Santulli, MD

Academic Editor

PLOS ONE

---

## [Editor Report · Acceptance letter]

8 May 2024

PONE-D-23-40628R1 

PLOS ONE

Dear Dr. Walkey, 

I'm pleased to inform you that your manuscript has been deemed suitable for publication in PLOS ONE. Congratulations! Your manuscript is now being handed over to our production team.

Kind regards, 

on behalf of

Professor Gaetano Santulli 

Academic Editor

PLOS ONE